# ICU-Acquired Pneumonia Is Associated with Poor Health Post-COVID-19 Syndrome

**DOI:** 10.3390/jcm11010224

**Published:** 2021-12-31

**Authors:** Ignacio Martin-Loeches, Anna Motos, Rosario Menéndez, Albert Gabarrús, Jessica González, Laia Fernández-Barat, Adrián Ceccato, Raquel Pérez-Arnal, Dario García-Gasulla, Ricard Ferrer, Jordi Riera, José Ángel Lorente, Óscar Peñuelas, Jesús F. Bermejo-Martin, David de Gonzalo-Calvo, Alejandro Rodríguez, Ferran Barbé, Luciano Aguilera, Rosario Amaya-Villar, Carme Barberà, José Barberán, Aaron Blandino Ortiz, Elena Bustamante-Munguira, Jesús Caballero, Cristina Carbajales, Nieves Carbonell, Mercedes Catalán-González, Cristóbal Galbán, Víctor D. Gumucio-Sanguino, Maria del Carmen de la Torre, Emili Díaz, Elena Gallego, José Luis García Garmendia, José Garnacho-Montero, José M. Gómez, Ruth Noemí Jorge García, Ana Loza-Vázquez, Judith Marín-Corral, Amalia Martínez de la Gándara, Ignacio Martínez Varela, Juan Lopez Messa, Guillermo M. Albaiceta, Mariana Andrea Novo, Yhivian Peñasco, Pilar Ricart, Luis Urrelo-Cerrón, Angel Sánchez-Miralles, Susana Sancho Chinesta, Lorenzo Socias, Jordi Solé-Violan, Luis Tamayo Lomas, Pablo Vidal, Antoni Torres

**Affiliations:** 1CIBER of Respiratory Diseases (CIBERES), Institute of Health Carlos III, 28029 Madrid, Spain; drmartinloeches@gmail.com (I.M.-L.); amotos@clinic.cat (A.M.); rosmenend@gmail.com (R.M.); gabarrus@clinic.cat (A.G.); lfernan1@clinic.cat (L.F.-B.); aceccato@clinic.cat (A.C.); r.ferrer@vhebron.net (R.F.); joriera@vhebron.net (J.R.); jose.angel.lorente@icloud.com (J.Á.L.); oscar.penuelasro@salud.madrid.org (Ó.P.); jfbermejo@saludcastillayleon.es (J.F.B.-M.); jsolvio@gobiernodecanarias.org (J.S.-V.); 2Pulmonary Department, Hospital Clinic, Universitat de Barcelona, IDIBAPS, 08036 Barcelona, Spain; 3Department of Intensive Care Medicine, St. James’s Hospital, Multidisciplinary Intensive Care Research Organization (MICRO), James’s Street, D08 NHY1 Dublin, Ireland; 4Pulmonary Department, University and Polytechnic Hospital La Fe, 46026 Valencia, Spain; 5Translational Research in Respiratory Medicine Group (TRRM), Lleida Biomedical Research Institute (IRBLleida), 25198 Lleida, Spain; jgonzalezgutierrez88@gmail.com (J.G.); dgonzalo@irblleida.cat (D.d.G.-C.); febarbe.lleida.ics@gencat.cat (F.B.); 6Pulmonary Department, Hospital Universitari Arnau de Vilanova and Santa Maria, 25198 Lleida, Spain; 7Barcelona Supercomputing Centre (BSC), 08034 Barcelona, Spain; raquel.perez@bsc.es (R.P.-A.); dario.garcia@bsc.es (D.G.-G.); 8Intensive Care Department, Vall d’Hebron Hospital Universitari, SODIR Research Group, Vall d’Hebron Institut de Recerca (VHIR), 08035 Barcelona, Spain; 9Hospital Universitario de Getafe, 28905 Madrid, Spain; 10Hospital Universitario Río Hortega de Valladolid, 47012 Valladolid, Spain; 11Instituto de Investigación Biomédica de Salamanca (IBSAL), Gerencia Regional de Salud de Castilla y León, 47007 Valladolid, Spain; 12Critical Care Department, Hospital Joan XXIII, 43005 Tarragona, Spain; ahr1161@yahoo.es; 13Anestesia, Reanimación y Terapia del Dolor, Hospital Universitario de Basurto, 48013 Bilbao, Spain; lucianojose.aguileracelorrio@osakidetza.eus; 14Intensive Care Clinical Unit, Hospital Universitario Virgen de Rocío, 41013 Sevilla, Spain; ramayavillar@gmail.com; 15Hospital Santa Maria, IRBLleida, 25198 Lleida, Spain; barberan60@gmail.com; 16Critical Care Department, Hospital Universitario HM Montepríncipe, Universidad San Pablo-CEU, 28660 Madrid, Spain; barberan@telefonica.net; 17Servicio de Medicina Intensiva, Hospital Universitario Ramón y Cajal, 28034 Madrid, Spain; ablandinoortiz@gmail.com; 18Department of Intensive Care Medicine, Hospital Clínico Universitario Valladolid, 47003 Valladolid, Spain; ebustamante@saludcastillayleon.es; 19Critical Care Department, Hospital Universitari Arnau de Vilanova, IRBLleida, 25198 Lleida, Spain; jcaballero.lleida.ics@gencat.cat; 20Hospital Álvaro Cunqueiro, 36213 Vigo, Spain; cristina.carbajales.perez@sergas.es; 21Intensive Care Unit, Hospital Clínico y Universitario de Valencia, 46010 Valencia, Spain; edurnecarbonell@yahoo.es; 22Department of Intensive Care Medicine, Hospital Universitario 12 de Octubre, 28041 Madrid, Spain; mmcges@gmail.com; 23Department of Medicine, CHUS, Complejo Hospitalario Universitario de Santiago, 15076 Santiago de Compostela, Spain; cristobal.galban.rodriguez@sergas.es; 24Department of Intensive Care, Hospital Universitari de Bellvitge, L’Hospitalet de Llobregat, 08907 Barcelona, Spain; vgumucio@bellvitgehospital.cat; 25Hospital de Mataró de Barcelona, 08301 Mataró, Spain; mctorre@csdm.cat; 26Department of Medicine, Universitat Autònoma de Barcelona (UAB), Critical Care Department, Corpo-Ració Sanitària Parc Taulí, Sabadell, 08208 Barcelona, Spain; emilio.diaz.santos@gmail.com; 27Unidad de Cuidados Intensivos, Hospital San Pedro de Alcántara, 10003 Cáceres, Spain; elenagallegocurto@gmail.com; 28Intensive Care Unit, Hospital San Juan de Dios del Aljarafe, 41930 Sevilla, Spain; joseluis.garciagarmendia@sjd.es; 29Intensive Care Clinical Unit, Hospital Universitario Virgen Macarena, 41009 Seville, Spain; jgarnachom@gmail.com; 30Hospital General Universitario Gregorio Marañón, 28009 Madrid, Spain; j.gomez@salud.madrid.org; 31Intensive Care Department, Hospital Nuestra Señora de Gracia, 50009 Zaragoza, Spain; ruthjorge@telefonica.net; 32Unidad de Medicina Intensiva, Hospital Universitario Virgen de Valme, 41014 Sevilla, Spain; aloza@telefonica.net; 33Critical Care Department, Hospital del Mar-IMIM, 08003 Barcelona, Spain; jmarincorral@gmail.com; 34Department of Intensive Medicine, Hospital Universitario Infanta Leonor, 28031 Madrid, Spain; am.gandara@hotmail.com; 35Critical Care Department, Hospital Universitario Lucus Augusti, 27003 Lugo, Spain; ignacioyag.martinez@gmail.com; 36Critical Care Department, Complejo Asistencial Universitario de Palencia, 34005 Palencia, Spain; jlomessa@gmail.com; 37Departamento de Biología Funcional, Instituto Universitario de Oncología del Principado de Asturias, Universidad de Oviedo, 33011 Oviedo, Spain; Guillermo.muniz@sespa.es; 38Instituto de Investigación Sanitaria del Principado de Asturias, Hospital Central de Asturias, 33011 Oviedo, Spain; 39Servei de Medicina Intensiva, Hospital Universitari Son Espases, Palma de Mallorca, 07120 Illes Balears, Spain; mariana.novo@ssib.es; 40Servicio de Medicina Intensiva, Hospital Universitario Marqués de Valdecilla, 39008 Santander, Spain; yhivian.penasco@scsalud.es; 41Servei de Medicina Intensiva, Hospital Universitari Germans Trias, 08916 Badalona, Spain; pricart.germanstrias@gencat.cat; 42Hospital Verge de la Cinta, 08916 Tortosa, Spain; urrece@gmail.com; 43Hospital de Sant Joan d’Alacant, 03550 Alacant, Spain; an.sanchezm1@coma.es; 44Servicio de Medicina Intensiva, Hospital Universitario y Politécnico La Fe, 46026 Valencia, Spain; sancho_sus@gva.es; 45Intensive Care Unit, Hospital Son Llàtzer, Palma de Mallorca, 07198 Illes Balears, Spain; lsocias@hsll.es; 46Critical Care Department, Hospital Dr. Negrín., 35019 Las Palmas de GC, Spain; 47Critical Care Department, Hospital Universitario Río Hortega de Valladolid, 47102 Valladolid, Spain; ltamayo@saludcastillayleon.es; 48Intensive Care Unit, Complexo Hospitalario Universitario de Ourense, 32005 Ourense, Spain; pablo.vidal.cortes@sergas.es

**Keywords:** COVID-19, CT abnormalities, ICU, lung function, SARS-CoV-2, sequelae, post-COVID

## Abstract

Background. Some patients previously presenting with COVID-19 have been reported to develop persistent COVID-19 symptoms. While this information has been adequately recognised and extensively published with respect to non-critically ill patients, less is known about the incidence and factors associated with the characteristics of persistent COVID-19. On the other hand, these patients very often have intensive care unit-acquired pneumonia (ICUAP). A second infectious hit after COVID increases the length of ICU stay and mechanical ventilation and could have an influence on poor health post-COVID 19 syndrome in ICU-discharged patients. Methods: This prospective, multicentre, and observational study was carrid out across 40 selected ICUs in Spain. Consecutive patients with COVID-19 requiring ICU admission were recruited and evaluated three months after hospital discharge. Results: A total of 1255 ICU patients were scheduled to be followed up at 3 months; however, the final cohort comprised 991 (78.9%) patients. A total of 315 patients developed ICUAP (97% of them had ventilated ICUAP). Patients requiring invasive mechanical ventilation had more persistent post-COVID-19 symptoms than those who did not require mechanical ventilation. Female sex, duration of ICU stay, development of ICUAP, and ARDS were independent factors for persistent poor health post-COVID-19. Conclusions: Persistent post-COVID-19 symptoms occurred in more than two-thirds of patients. Female sex, duration of ICU stay, development of ICUAP, and ARDS all comprised independent factors for persistent poor health post-COVID-19. Prevention of ICUAP could have beneficial effects in poor health post-COVID-19.

## 1. Introduction

Clinical presentation of severe acute respiratory syndrome coronavirus 2 (SARS-CoV-2) infection ranges from mild to severe [1]. Disease severity, including refractory acute respiratory failure (ARF) and acute respiratory distress syndrome (ARDS), may require admission to an intensive care unit (ICU) [2]. Patients often need invasive mechanical ventilation and can develop multiorgan failure [3]. Duration of ICU stay is long among survivors, and mortality can be high in patients with ARDS, reaching between 40 and 50% [4]. On the other hand, ICU-admitted patients, especially those requiring mechanical ventilation, have a high incidence of other infection complications such as ICU-acquired pneumonia (ICUAP), which could have a role in the persistence of symptoms after discharge [5]. ICU-acquired pneumonia (ICUAP) is the most common hospital-acquired infection in the ICU. This infection encompasses two different entities: ventilator-associated pneumonia (VAP) and hospital-acquired pneumonia (HAP) in non-intubated patients during their ICU stay [6]. Additionally, although often sufficiently recovered for hospital discharge, patients with either mild or severe disease are at risk of developing a condition known as persistent poor health post-COVID-19, post-COVID syndrome, or long COVID [7,8,9].

Currently, no consensus definition for the symptoms of poor health post-COVID-19 exist; however, the most common symptoms include fatigue, shortness of breath, weakness, and asthenia [10]. Some publications have reported a high incidence of persistent poor health post-COVID-19. Specifically, fatigue and dyspnoea are the most frequent symptoms, and their incidence and intensity are not correlated with initial disease severity [11]. With respect to information regarding persistent poor health post-COVID-19 in patients that have survived a stay in the ICU, little remains known [12,13,14]. Very importantly, factors associated with those present in the acute period of the disease are not well known. As pointed out in a recent position manuscript, the recognition of factors associated with the acute period is a research priority for understanding the long-term sequalae of COVID-19 [15].

In the present manuscript, we analysed poor health post-COVID-19 in the initial cases of hospitalised patients with COVID-19 at 3-month follow-up after hospital discharge. We hypothesised that critically ill patients would present with high, persistent post-COVID-19 symptoms and significant abnormalities in both lung function tests and radiology. Herein, we summarise the poor health post-COVID-19, functional respiratory parameters, and radiological features of patients discharged from the ICU at 3-month follow-up. The main objective of this study was to determine the factors of the acute period associated with poor health post-COVID-19 in ICU survivors at 3-month follow-up following hospital discharge and identify the factors associated with poor recovery. We also aim to analyse lung function and radiologic abnormalities in critically ill patients after hospital discharge.

## 2. Material and Methods

### 2.1. Study Design and Population

We carried out a prospective, multicentre and observational cohort study at 40 selected ICUs in Spain with critically ill patients initially admitted from 16 February 2020 until 1 January 2021. We consecutively recruited patients with COVID-19 requiring ICU admission and performed a follow-up at three months after hospital discharge. This study is a pre-planned analysis of the ongoing multicentre study called CIBERESUCICOVID (ClinicalTrials.gov Identifier: NCT04457505). We then asked staff from each centre to prospectively obtain data for ICU-admitted patients aged 18 years or older with positive polymerase chain reactions (PCR) for SARS-CoV-2. Re-admitted patients and previously tracheostomised patients were not included. The study received approval by the institution’s Internal Review Board (Comité Ètic d’Investigació Clínica, registry number HCB/2020/0370). We obtained informed consent for most patients by using emergency consent mechanisms in accordance with ethics approval guidelines for the study. Further participating centres either received ethics approval from their institutions or had waived ethics approval. Finally, we de-identified all clinical data to allow for the waiver of informed consent.

### 2.2. Data Collection

Recorded data included demographic characteristics, comorbidities, time course of illness, treatments administered, laboratory and microbiologic data, radiologic findings on chest X-rays, CT scans, ventilatory parameters in patients with invasive mechanical ventilation, complications during ICU stay, and outcomes. We determined disease severity and assessed organ failure using the Sequential Organ Failure Assessment (SOFA) score, calculating both within the first day of ICU admission [16]. Ventilatory management strategies were not standardised among centres and were left to the discretion of the attending clinician, based on National Ministry of Health recommendations, and supported by international guidelines. We defined ICU-acquired pneumonia (ICUAP) as pneumonia developing in patients in the ICU for ≥48 h. Basis of ICUAP diagnosis comprised new or progressive radiologic pulmonary infiltrates together with at least two of the subsequent characteristics: temperature > 38 °C or < 36 °C; leucocytosis > 12,000/mm^3^ or leucopoenia < 4000/mm^3^; or purulent respiratory secretion [5,17]. We confirmed an ARDS diagnosis using the Berlin definition [18].

### 2.3. Procedures

Poor health post-COVID-19 was defined by a voluntary report of any of the following symptoms after COVID infection: dyspnoea, weakness, asthenia, myalgia, cough, numbness, headache, anosmia and ageusia. We provided electronic case report forms using a secure website. For all patients, we recorded demographic characteristics, duration of ICU and hospital stays, the McCabe classification of comorbidities and likelihood of survival (likely to survive 5 years, 1–5 years (ultimately fatal), or <1 year (rapidly fatal)), the SOFA score to predict hospital mortality, and outcome (ICU mortality). We calculated static compliance of the respiratory system as tidal volume/(end-inspiratory plateau pressure–total PEEP). Chest CT scans and CT pulmonary angiograms were obtained when clinically indicated and technically feasible. All the radiological interpretation was performed by an independent certified radiologist in all cases, as per current clinical practice in the participating hospitals. The Modified Medical Research Council (mMRC) dyspnoea scale grades the impact of dyspnoea on daily activities throughout the prior seven days and thereby quantifies the disability or physical limitations associated with dyspnoea [19]. Finally, to aid analysis, we clustered patients into groups according to clinical resolution.

### 2.4. Outcomes

The primary outcome of our study was to determine incidence and the factors associated with poor health post-COVID-19 in critically ill patients per clinical presentation at three months of hospital discharge. The secondary outcome included determining associations between poor health post-COVID-19 symptoms and abnormalities in lung function tests, radiologic characteristics, and laboratory parameters.

### 2.5. Statistical Analysis

The study sample size was not formally calculated but instead based on the nature of the study and pre-planned dates. We used SPSS (version 20) for data analysis. All *p* values were two-tailed. We considered differences as significant if *p* was less than 0.05. We reported categorical variables as numbers and frequencies (%), normally distributed continuous variables as means (standard deviation (SD)) and skewed continuous variables as medians ((interquartile range (IQR)). We performed both χ2 tests or Fisher’s exact tests to compare qualitative variables and Student’s t tests and ANOVAs or Mann–Whitney U and non-parametric Kruskal–Wallis tests to compare normally distributed or skewed continuous variables, whenever appropriate. We undertook univariate analyses of predictors of post-COVID syndrome, using the χ2 test and Student’s t test. To explore the factors associated with post-COVID syndrome, a generalized linear model was used, defined by a binomial probability distribution and a log link function. The following variables were included in the multivariable model based on clinical relevance only: sex, SOFA, cirrhosis, non-invasive mechanical ventilation, invasive mechanical ventilation, ICU length of stay, corticosteroids, ICUAP, and ARDS. Relative risks (RRs) and their 95% confidence intervals were calculated. We performed a mixed-effects model [20] for sensitivity analysis, as defined by a Poisson probability distribution and a log link function, with centres as a random effect, and with an unstructured covariance matrix.

## 3. Results

### 3.1. Definition of the Population

We monitored patients after hospital discharge for a median of 77 (57–99) days. A total of 1255 patients were scheduled to be followed up for three months. However, we could not perform a 3-month follow-up of 264 patients. Patients did not show up to their appointments despite multiple attempts. The final cohort comprised 991 (78.9%) patients. A total of 731 (73.8%) patients developed persistent post-COVID-19 symptoms. A flow chart with the percentage of patients with persistent post-COVID-19 symptoms is displayed in Figure 1. Patient characteristics are shown in Table 1.

We performed the mMRC dyspnoea scale in 402 patients, of whom 59% were grade 1, 28.4% grade 2, 10% grade 3, and 2.2% grade 4. Physical examination of patients revealed crackles in 8.2% (n = 82) and showed a trend of its presence in patients with persistent post-COVID-19 symptoms (9.9% vs. 5.8%, *p* = 0.05). After discharge, 33 (3.2%) patients presented with infectious complications, 172 (15.1%) attended the emergency department, and 63 (5%) were readmitted to hospital. No significant differences were observed in patients with or without persistent post-COVID-19 symptoms with respect to follow-up clinic visits (63.5% vs. 59.6%, *p* 0.3). Patients with persistent post-COVID-19 symptoms required additional respiratory therapy after hospital discharge (Appendix A).

### 3.2. Clinical Features and Associations with Functional (Lung Function Tests), Imaging and Laboratory Results at 3-Month Follow-Up

We performed a radiologic work-up in 471 (47.5%) patients. A similar number of patients with and without persistent post-COVID-19 symptoms underwent either chest radiography (51.4% vs. 48.1%, *p* 0.3) or chest CT scans (40.5% vs. 45.9%, *p* = 0.1). Persistently abnormal chest X-rays were observed in 21.4% of patients. Patients with persistent post-COVID-19 symptoms more frequently had abnormal X-rays and chest CT scans showing diffuse interstitial lung patterns than those without such symptoms. Patients who required invasive mechanical ventilation more often presented interstitial lung disease patterns in chest CT scans. Further details are shown in Table 2.

We performed a pulmonary function test (PFT) in 535 (53.9%) patients and diffusing capacity of the lungs for carbon monoxide (DLCO) in 464 (46.8%). PFTs (57.1% vs. 49%, *p* = 0.03) were performed more often in patients with persistent post-COVID-19 symptoms, while there were no significant differences in performing DLCO between patients with or without persistent post-COVID-19 symptoms (50.7% vs. 55.3%, *p* = 0.2). A total of 665 (67.4%) patients underwent invasive mechanical ventilation. Patients with invasive mechanical ventilation had a higher percentage of persistent post-COVID-19 symptoms than those without mechanical ventilation (70.2% vs. 59.6%, *p* = 0.003). Table 3 shows invasive mechanical ventilator and oxygenation parameters obtained sequentially at days 1 and 3 after the initiation of invasive mechanical ventilation. Only pulmonary compliance calculated at the time of initiation of mechanical ventilation and the value of PaO2/FiO2 at day 3 were significantly worse in patients with persistent post-COVID-19 symptoms.

Laboratory data at day 1 of ICU admission are detailed in Table 4. No significant differences were observed in most parameters; however, fibrinogen was significantly higher in patients with persistent post-COVID-19 symptoms.

### 3.3. Factors Associated with Persistent Post-COVID-19 Symptoms

Table 5 shows factors associated with persistent post-COVID-19 symptoms. Many variables showed a significant association in the univariate analysis. A multivariable analysis found that four factors were associated with persistent post-COVID-19 symptoms: female sex, duration of ICU stay, the development of ICUAP, and ARDS. In terms of percent change, the RR of persistent post-COVID-19 symptoms in females is 1.69, indicating that females are about 69% more likely to develop persistent post-COVID-19 than males; the RR of persistent post-COVID-19 symptoms in patients with prolonged ICU stay (length of stay > 14 days) is 1.54, indicating that patients with prolonged ICU stay are about 54% more likely to develop persistent post-COVID-19 than patients with short ICU stay (length of stay ≤ 14 days); the RR of persistent post-COVID-19 symptoms for the onset of IUCAP is 1.88, indicating that the risk of persistent post-COVID-19 symptoms for the onset of ICUAP are about 88% more likely to develop persistent post-COVID-19 than for no onset of ICUAP; and the RR of persistent post-COVID-19 symptoms for the onset of ARDS is 1.41, indicating that the risk of persistent post-COVID-19 symptoms for the onset of ARDS are about 41% more likely to develop persistent post-COVID-19 than for no onset of ARDS. A sensitivity analysis introducing the centre variable as a random effect in the mixed-effects multivariable model yielded similar results.

Among the 282 patients with confirmed pathogen the mean period of days from ICU admission was 12 days (7–21 days). Among the 258 patients with confirmed pathogen, 220 were ventilated. 44 (20%) had early VAP and 176 (80%) had late VAP. The most pathogens isolated were Gram negatives, but S. aureus was present in 46 (16%) cases. 40 patients had polymicrobial pneumonia.

## 4. Discussion

The present study analysed a multicentre cohort of patients admitted to the ICU with COVID-19. Few data are currently available on the follow-up of survivor patients with COVID-19 after discharge. Interestingly, we found that more than 70% of patients discharged from the ICU had persistent post-COVID-19 symptoms after three months had passed since hospital discharge; however, the hospital readmission rate within this period remained low, and only 15% needed to visit the emergency department. There was also a poor correlation between abnormal radiologic findings and persistent post-COVID-19 symptoms in critically ill patients after three months had passed since hospital discharge. Factors associated with persistent post-COVID-19 symptoms included female sex, duration of ICU stay, development of ICUAP and ARDS.

With respect to the burden of follow-up on the healthcare system, X-rays and CT scans were performed in 4 of 10 ICU-admitted patients at three months after hospital discharge. As expected, normal X-rays were frequent in patients with good clinical resolution; interstitial patterns were more often seen in chest CT scans of patients with persistent post-COVID-19 symptoms. Recently, patients with previously undiagnosed fibrotic lung abnormalities have been reported to face the possibility of ARDS onset [21]. In a cohort of 114 survivors of severe COVID-19 monitored for six months, chest CT scans revealed fibrotic-like changes in the lungs in more than one-third of cases [22]. In contrast, only 2.5% of our cohort presented with an interstitial lung pattern. In our view, an important issue is to determine patients’ need for resources after hospital discharge. More than 15% of our cohort of patients discharged from the ICU required oxygen, yet at the time of writing this manuscript, only a minority (5%) continued receiving supplementary oxygen at home. Interestingly, nebulizers were less frequently used than supplementary oxygen after hospital discharge. To the best of our knowledge, this is a novel finding. Few data are, however, available regarding additional therapy in patients with COVID-19 discharged from the ICU; this is a point that warrants further exploration. Investigators Banerjee et al. [23] followed 621 discharged patients receiving oxygen at home and reported a 30-day hospital readmission rate of 8.5%. Readmission rate in our cohort was much lower than that in that study (3.1%), although more patients (15%) visited the emergency department after discharge.

The symptoms most frequently observed in patients with post-COVID-19 included dyspnoea, asthenia, and weakness. After analysis of the therapy provided, we found that oxygen therapy was provided significantly more often to patients with persistent post-COVID-19 symptoms. In a previous study in China, patients monitored for three months after hospital discharge presented with considerable radiologic and physiologic abnormalities [24]. In another Chinese study [25] including over 1000 patients, survivors of COVID-19 presented with fatigue, sleep difficulties and anxiety or depression at 6-month follow-up. However, as in the previous study, no detailed data about the functional status of the patients were provided. In our study, we did not perform any analysis related to depression or anxiety.

A strength of our research includes the prospective follow-up of a detailed list of lung function parameters. Soriano et al. [26] recently wrote an editorial suggesting more studies be done in clinical research assessing the post-COVID-19 condition. Our study integrates an extremely sizeable cohort and evaluates a relevant subgroup of the population, i.e., critically ill patients. A French study found that patients with COVID-19 had some symptoms not previously present before their disease [27]. These findings complement our report, as most of the patients included in our cohort were in critically ill condition. Moreover, in our cohort, ICU-admitted patients with no clinical resolution had worse forced expiratory volume in the first second (FEV1) than those who did not present with persistent post-COVID-19 symptoms. Some studies—the majority from China and some from Europe, with limited patient samples—have also found substantial differences in PFT; however, most patients included were not critically ill [28,29]. In addition to FEV1, FEV1/FVC and DLCO presented significant differences. In a study performed in Sweden, investigators Ekbom et al. [30] found that over half of patients with COVID-19 treated in the ICU had impaired lung function during follow-up, suggesting further follow-up studies including DLCO. In their cohort, a mean DLCO of 62% was reported, as predicted among those with abnormal DLCO. These figures are like ours. We observed a significant correlation between abnormal DLCO and poor health post-COVID-19 in our cohort. The presence of decreased DLCO might reflect microvascular or alveolar capillary damage and be expected in patients with no clinical resolution [31]. Very little is known about the pathophysiology of poor health post-COVID-19. COVID-19 causes lung damage due to a marked inflammatory response to the virus. As is known, the disease may damage endothelial cells in the lung parenchyma. Therefore, identifying pathways may prove as a key point in determining this damage. Ward et al. [32] found that increased plasma levels of von Willebrand factor antigen (VWF:Ag) and pro-coagulant factor VIII (FVIII) were seen in patients with SARS-CoV-2 infection. In our cohort, we found only elevated levels of fibrinogen in patients with persistent post-COVID-19 symptoms; this observation could help determine the role of endothelial activation in pathophysiology of the disease.

Additionally, we aimed to determine an association between ventilatory parameters in patients with invasive mechanical ventilation and persistent post-COVID-19 symptoms at the 3-month follow-up after hospital discharge. Our assessment of different respiratory parameters such as PaO2/FiO2 ratio and pCO2 showed a significant correlation across two variables: compliance at the time of intubation and PCO2 at day 3. Both parameters clearly reflect the damage caused to the respiratory system by a COVID-19 infection. It is interesting that compliance, albeit not the PaO2/FiO2 ratio, was a predictor of persistent post-COVID-19 symptoms. A recent study found that median time to intubation was twice as long in the very-low compliance group than in the low-normal compliance group [33]. Reported higher levels of PaCO2 in patients in the very-low lung compliance group in that study correlated strongly with our findings. Furthermore, some authors [34] have suggested that compliance in COVID-19-related ARDS is higher in non-COVID-19-related ARDS; our findings could be explained by the fact that the patients with low compliance were those with more severe ARDS. Recently, Gonzalez et al. found that abnormal results were present in CT scans of more than two-thirds of patients with COVID-19-related ARDS [35].

A factor associated with poor recovery was female sex. This is an intriguing finding, given that male patients are more widely reported to be admitted to the ICU [36,37] for COVID-19. Furthermore, a systematic review found that COVID-19 may be associated with worse outcomes in males than in females [38]. While most ICU-admitted patients in our cohort were male (67%), more female patients had persistent post-COVID-19 symptoms at 3-month follow-up. The PHOSP-COVID study conducted in the United Kingdom observed that 70% had not fully recovered after a mean follow-up period of five months following hospital discharge, with the percentage in women being greater than that in men [39]. Another group who presented with poor recovery were those with ARDS. Some previous non-COVID-19 studies have found that ARDS patients who survive ARDS had similar substantial reduction in health-related quality of life compared to other critical care patients without ARDS [40]. In COVID-19, a small study found that a majority of invasively mechanically ventilated survivors at 3 months after hospital discharge had abnormal pulmonary function tests and residual changes on CT scans [41].

Lastly, the development of a ICUAP during the ICU stay was found to be an independent factor associated with poor recovery at 3-month follow-up. This is an especially important finding given the high incidence of nosocomial pneumonia in critically ill patients, especially in those that needed invasive mechanical ventilation. This finding stresses the importance of the prevention ICUAP in COVID-19 critically ill patients. The hypothesis behind this finding is that a second hit (ICUAP) after COVID-19 increased lung damage and consequently increased the risk of respiratory symptoms persistence at follow-up. Some multicentre, European manuscripts suggest that ventilator-associated lower respiratory tract infections (VA-LRTI) were more frequent in patients with COVID-19 than in patients admitted to the ICU with another virus (influenza) or in patients without viral infections [42]. To the best of our knowledge, this finding has not been reported elsewhere, and other studies should be carried out to confirm or refute it.

Our study has several limitations. Samples obtained from the centres may not be representative, given that the hospital units selected all had the research resources necessary to participate. We included an acceptable number of variables for follow-up analysis; however, certain functional tests were not recorded, including the six-min walk test (6MWT), an excellent tool for assessing sub-maximal exercise aerobic capacity and endurance. We preferred to determine PFT and imaging, as detailed extensively in this manuscript, and felt that the 6MWT might not be reproducible as a measure for oxygen desaturation. Another potential limitation was related to symptom collection. The objective of the study was to obtain a voluntary report of symptoms after the onset of COVID and some patients might have had these symptoms before contracting COVID for various reasons. Moreover, the database did not capture the frequency of headaches and/or the location of numbness. A strength of this manuscript is the availability of many data points from the acute period and including data of day 1 and day 3 to determine factors associated with long-COVID 19 syndrome.

## 5. Conclusions

In conclusion, we evaluated many critically ill patients with COVID-19 after three months had passed following hospital discharge. Persistent post-COVID-19 symptoms occurred in more than two-thirds of patients; however, the hospital readmission rate remained low. There was no clinical association between such symptoms and persistently abnormal chest X-rays. Additionally, more than 10% of these patients still required oxygen at home. Female sex, duration of ICU stay, development of ICUAP, and ARDS were independent factors associated with persistent post-COVID-19 symptoms in critically ill patients at 3-month follow-up. Prevention of ICUAP could have beneficial effects in poor health post-COVID 19 syndrome.

## Figures and Tables

**Figure 1 jcm-11-00224-f001:**
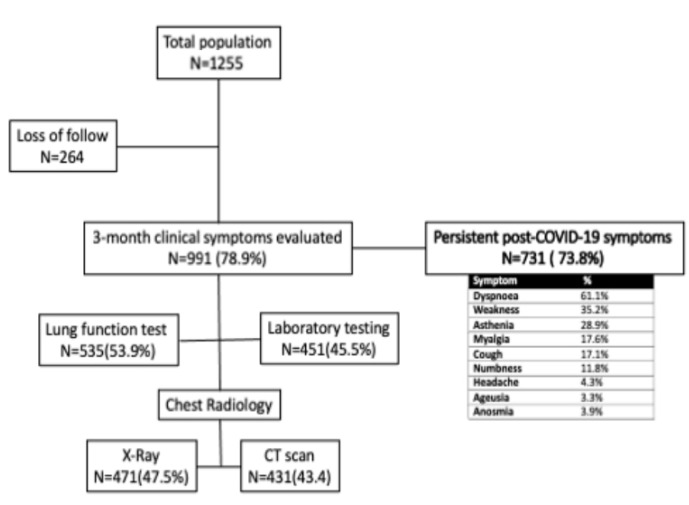
Flow chart and the percentage of patients with persistent port-COVID-19 symptoms.

**Table 1 jcm-11-00224-t001:** Patient characteristics of the population enrolled in the study based on persistent post-COVID-19 symptoms at 3-month follow-up.

Demographic Item	*n*
Age *, mean (SD), years	58.5 (SD11.9)
Sex (Female), *n* (%)	326 (32.9)
SOFA *, mean (SD)	5.1 (SD 3.0)
CHF, *n* (%)	92 (9.3)
Hypertension, *n* (%)	441 (44.5)
COPD, *n* (%)	81 (8.2)
Asthma, *n* (%)	65 (6.6)
CKD), *n* (%)	53 (5.4)
Cirrhosis, *n* (%)	13 (1.3)
Mild liver failure, *n* (%)	19 (1.9)
Neurologic, *n* (%)	48 (4.8)
Dementia, *n* (%)	3 (0.3)
Autoimmune, *n* (%)	56 (5.7)
Gastrointestinal, *n* (%)	80 (8.1)
Endocrine, *n* (%)	75 (7.6)
Obesity (BMI >30 kg/m^2^, *n* (%)	391 (39.5)
Diabetes Mellitus, *n* (%)	186 (18.8)
Haematologic disease, *n* (%)	52 (5.3)
Solid cancer, *n* (%)	29 (2.9)
Transplant, *n* (%)	4 (0.4)
HIV, *n* (%)	8 (0.6)
Smoking, *n* (%)	42 (4.2)
Alcohol), *n* (%)	28 (2.8)
Oxygen requirement, *n* (%)	981 (99.6)
NIMV, *n* (%)	348 (35.1)
iMV, *n* (%)	665 (67.4)
Prone, *n* (%)	567 (57.4)
Tracheostomy, *n* (%)	312 (31.6)
ICU length of stay, mean (SD), days	20.1 (18.2)
ECMO, *n* (%)	18 (1.8)
CRRT, *n* (%)	54 (5.5)
Shock, *n* (%)	596 (60.4)
NMB, *n* (%)	554 (56.2)
Corticosteroids, *n* (%)	750 (76.7)
CPR, *n* (%)	7 (0.7)
ICUAP, *n* (%)	260 (26.4)
ARDS, *n* (%)	739 (74.9)
Pneumothoraz, *n* (%)	45 (4.6)
COP, *n* (%)	46 (4.7)
PE, *n* (%)	97 (9.8)
Delirium, *n* (%)	234 (23.8)

Abbreviations: SOFA: Sequential Organ Failure Assessment. CHF: congestive heart failure. COPD: Chronic obstructive pulmonary disease. CKD: Chronic kidney disease. BMI: Body mass index. HIV: Human immunodeficiency virus. NIMV: Non-invasive mechanical ventilation. iMV: Invasive mechanical ventilation. ICU: Intensive care unit. ECMO: Extracorporeal membrane oxygenation. CRRT: Continuous renal replacement therapy. NMB: Neuromuscular blockade. CPR: Cardiopulmonary resuscitation. ICUAP: Intensive care unit-acquired pneumonia. ARDS: Acute respiratory distress syndrome. COP: Cryptogenic organizing pneumonia. * Mean and SD.

**Table 2 jcm-11-00224-t002:** Chest imaging and lung function tests in patients with persistent post-COVID-19 symptoms and in those who underwent invasive mechanical ventilation at 3-month follow-up.

	Post-COVID			Invasive Mechanical Ventilation		
	Yes	No	*p*-Value	Yes	No	*p*-Value
Chest X-ray, *n* (%)						
Abnormal	597 (81.7)	182 (70)	<0.01	540 (73.0)	235 (81.2)	0.004
ILD	2 (0.8)	5 (0.7)	0.5	5 (0.8)	2 (0.6)	0.9
Persistent infiltrates	34 (13.1)	130 (17.8)	0.08	118 (17.7)	45 (14)	0.1
Fibrotic tract	12 (4.6)	54 (7.4)	0.1	50 (5.8)	19 (4.9)	0.5
Emphysema	0	3 (0.4)	0.5	3 (0.5)	0	0.5
CT scan, *n* (%)						
Abnormal	685 (93.7)	238 (91.5)	0.2	619 (93.1)	300 (93.2)	0.9
ILD	0	18 (2.5)	0.006	6 (1.9)	12 (1.8)	0.9
Persistent infiltrates	43 (16.5)	118 (16.1)	0.9	120 (18)	40 (12.4)	0.02
Fibrotic	27 (10.4)	88 (12)	0.5	81 (12.2)	34 (10.4)	0.5
Emphysema	4 (1.5)	23 (3.1)	0.2	20 (3)	7 (2.2)	0.5
PE, *n* (%)	2 (0.8)	11 (1.5)	0.2	12 (1.4)	1 (0.3)	0.07
PFTs, *n* (%)						
FEV1	94.2 (18.2)	87.5 (18.1)	<0.01	89.5 (19.2)	88.1 (17.1)	0.4
FEV1/FVC	91.4 (16.3)	97.2 (14.6)	<0.01	95.5 (14.8)	96.5 (15.8)	0.5
DLCO (mL/min/mm Hg)	78.3 (17.4)	67.1 (17.7)	<0.01	67.1 (18.2)	74.2 (17.4)	<0.01
DLCO 80	205 (78.8)	459 (62.8)	<0.01	441 (66.3)	220 (68.3)	0.5

Data are mean (SD) or number of patients (%). Abbreviations: SD: Standard deviation. CT: Computed tomography. ILD: Diffuse interstitial lung disease. PE: Pulmonary embolism. LFT: Lung function test. FEV1: Forced expiratory volume in the first second. FVC: Forced vital capacity. DLCO: Diffusing capacity for carbon monoxide.

**Table 3 jcm-11-00224-t003:** Respiratory parameters in patients with invasive mechanical ventilation based on persistent post-COVID-19 symptoms at 3-month follow-up.

		Post-COVID		
Variable	Time-Point	No	Yes	*p*-Value
PaO2/FiO2, mean (SD)	Intubation	130.5 (73.4)	131.6 (74.9)	0.8
	Day 3	192.0 (85.7)	180.0 (75.0)	0.1
	Change from intubation	66.3 (100.8)	49.0 (111.7)	0.1
PaCO2, mean (SD), mmHg	Intubation	39.4 (9.5)	40.4 (10.6)	0.2
	Day 3	43.0 (9.4)	44.8 (9.8)	0.03
	Change from intubation	3.4 (11.5)	3.4 (12.2)	1.0
VT admission, mean (SD), mL	Intubation	7.1 (1.2)	6.9 (3.0)	0.5
	Day 3	7.2 (1.2)	7.1 (3.1)	0.7
	Change from intubation	0.2 (1.0)	0.3 (1.6)	0.6
PEEP, mean (SD), cmH2O	Intubation	12.5 (2.5)	12.4 (2.5)	0.7
	Day 3	12.0 (2.7)	12.1 (2.7)	0.5
	Change from intubation	−0.6 (2.8)	-0.4 (2.9)	0.6
ΔP, mean (SD)	Intubation	10.8 (4.6)	11.8 (4.7)	0.1
	Day 3	10.7 (4.2)	11.8 (4.8)	0.2
	Change from intubation	−0.6 (3.0)	−0.1 (5.1)	0.7
Compliance, mean (SD)	Intubation	66.3 (86.4)	47.9 (35.9)	0.02
	Day 3	42.6 (79.0)	46.0 (29.0)	0.6
	Change from intubation	−26.1 (127.4)	0.2 (43.0)	0.1
VR, mean (SD)	Intubation	1.7 (0.5)	1.7 (0.5)	0.7
	Day 3	1.8 (0.5)	1.8 (0.5)	0.8
	Change from intubation	0.2 (0.5)	0.2 (0.6)	0.7

Data are mean (SD). Abbreviations: VT: Tidal volume. ΔP: Driving pressure PEEP: Positive end-expiratory pressure. VR: Ventilatory ratio. The delta measurements were computed as the difference in amplitude between day 3 of initiation of invasive mechanical ventilation and day 1 of initiation of invasive mechanical ventilation.

**Table 4 jcm-11-00224-t004:** Laboratory parameters in patients with invasive mechanical ventilation based on persistent post-COVID-19 symptoms at 3-month follow-up.

	Post-COVID		
	No	Yes	*p*-Value
	Value (SD)	Mean	
Haemoglobin, mean (SD), g/dL	14.7 (11.3)	13.9 (1.5)	0.3
WCC, mean (SD), x 109/L	7.3 (2.2)	6.81 (2.3)	0.06
Lymphocytes, mean (SD), × 109/L	2.2 (0.9)	2.3 (1.4)	0.6
Neutrophiles, mean (SD), × 109/L	4.4 (3.6)	3.7 (1.9)	0.09
Monocytes, mean (SD), × 109/L	0.6 (0.5)	0.5 (0.2)	0.2
Haematocrit, mean (SD), × 109/L	41.4 (5.2)	42.1 (4.6)	0.1
Platelets, mean (SD), × 109/L	259.3 (73.5)	247.4 (88.7)	0.2
Prothrombin time, mean (SD), sec	11.9 (6.8)	11.8 (4.1)	0.9
INR, mean (SD), IU	1.3 (0.6)	1.1 (0.3)	0.5
D-dimer, mean (SD), ng/mL	410.2 (445.1)	433.2 (393.4)	0.6
Fibrinogen, mean (SD), ng/mL	271.2 (249.7)	358.8 (192.5)	<0.01
CRP, mean (SD), mg/L	6.3 (0.4)	6.9 (0.6)	0.7
AST, mean (SD), IU/L	22.3 (13.4)	23.1 (20.2)	0.6
ALT, mean (SD), IU/L	24.8 (22.4)	24.1 (14.3)	0.6
GGT, mean (SD), IU/L	49.4 (5.6)	35.4 (5.7)	0.2
Urea, mean (SD), mg/dL	5.9 (1.8)	5.8 (1.2)	0.9
Creatinine (mg/dL	0.9 (0.8)	0.8 (0.4)	0.4
CK, mean (SD), IU/L	78.2 (6.7)	88.8 (9.9)	0.4
LDH, mean (SD), IU/L	250.4 (104.2)	235.7 (90.1)	0.1

Data are mean (SD). Abbreviations: L: Litres. Mg: milligrams. WCC: White blood cells. INR: international normalised ratio. Sec: seconds. IU: International Units. CRP: C-reactive protein. AST: Aspartate Transaminase. ALT: Alanine Aminotransferase. GGT: Gamma Glutamyl Transferase. CK: Creatine kinase. LDH: Lactate dehydrogenase (LDH).

**Table 5 jcm-11-00224-t005:** Univariate and multivariate analyses of factor associated with persistent post-COVID-19 symptoms at 3-month follow-up.

	Univariate (*n* = 991)			Multivariable (*n* = 620)	
	Post-COVID			RR (95% CI)	*p*-Value
	No	Yes	*p*-value		
Age, mean (SD), years	57.85 (12.9)	58.74 (11.5)	0.3		
Female sex, *n* (%)	66 (25.4)	260 (35.6)	<0.01	1.69 (1.23–2.32)	<0.01
SOFA, mean (SD)	4.6 (2.8)	5.2 (3.1)	0.01		
SOFA > 4, *n* (%)	65 (39.6)	244 (51.6)	<0.01	1.23 (0.90–1.67)	0.2
CHF, *n* (%)	28 (10.8)	64 (8.8)	0.3		
Hypertension, *n* (%)	115 (44.2)	326 (44.7)	0.9		
COPD, *n* (%)	18 (6.9)	63 (8.6)	0.4		
Asthma, *n* (%)	21 (8.1)	44 (6)	0.2		
CKD, *n* (%)	10 (3.8)	43 (5.9)	0.2		
Cirrhosis, *n* (%)	0	13 (1.8)	0.02	-*	-
Mild liver failure, *n* (%)	5 (1.9)	14 (1.9)	0.9		
Neurological, *n* (%)	9 (3.5)	39 (5.3)	0.3		
Dementia, *n* (%)	1 (0.4)	2 (0.3)	0.7		
Autoimmune, *n* (%)	11 (4.2)	45 (6.2)	0.2		
Gastrointestinal, *n* (%)	15 (5.8)	65 (8.9)	0.1		
Endocrine, *n* (%)	17 (6.5)	58 (7.9)	0.4		
Obesity (BMI >30 kg/m^2^), *n* (%)	98 (37.7)	293 (40.1)	0.5		
Diabetes Mellitus, *n* (%)	46 (17.7)	140 (19.2)	0.6		
Haematological disease, *n* (%)	13 (5)	39 (5.3)	0.8		
Solid cancer, *n* (%)	8 (3.1)	21 (2.9)	0.8		
Transplant, *n* (%)	2 (0.8)	9 (1.2)	0.7		
HIV, *n* (%)	0	4 (0.5)	0.5		
Oxygen requirement, *n* (%)	248 (99.2)	723 (99.7)	0.2		
NIV, *n* (%)	68 (26.5)	280 (38.9)	<0.01	1.01 (0.76–1.34)	0.9
iMV, *n* (%)	155 (59.6)	510 (70.2)	<0.01	0.93 (0.68–1.27)	0.6
Prone, *n* (%)	136 (52.3)	431 (59.4)	0.04		
Tracheostomy, *n* (%)	52 (20)	260 (35.8)	<0.01		
ICU length of stay, mean (SD), days	14.3 (12.2)	22.2 (18.2)	<0.01		
ICU length of stay >14 days, *n* (%)	91 (35)	382 (52.3)	<0.01	1.54 (1.11–2.14)	<0.01
ECMO, *n* (%)	52 (2 (0.8)	16 (2.2)	0.1		
CRRT, *n* (%)	7 (2.7)	47 (6.5)	0.02		
Shock, *n* (%)	132 (51.6)	46 (64.3)	<0.01		
NMB, *n* (%)	124 (48.2)	430 (59.6)	<0.01		
Corticosteroids, *n* (%)	207 (80.5)	543 (75.3)	0.1	0.75 (0.54–1.05)	0.1
CPR, *n* (%)	2 (0.8)	5 (0.7)	0.9		
ICUAP, *n* (%)	41 (15.9)	219 (30.2)	<0.01	1.88 (1.22–2.90)	<0.01
ARDS, *n* (%)	175 (68.4)	564 (77.6)	<0.01	1.41 (1.08–1.83)	0.01
NTX, *n* (%)	9 (3.5)	36 (5.0)	0.4		
COP, *n* (%)	11 (4.3)	35 (4.9)	0.9		
PE, *n* (%)	33 (13)	64 (9)	0.08		
Delirium, *n* (%)	52 (20.1)	182 (25.1)	0.1		

Data are mean (SD) or number of patients (%) for the univariate analysis and estimated RRs (95% CIs) of the explanatory variables in the persistent post-COVID-19 symptoms group for the multivariable analysis. Multivariable model is generalized linear model, considering a binomial probability distribution and a log link function. The RR represents the risk that the presence of persistent post-COVID-19 symptoms will occur given exposure of the explanatory variable, compared to the risk of the outcome occurring in the absence of that exposure. The *p*-value for the multivariable analysis is based on the null hypothesis that all RRs relating to an explanatory variable equal unity (no effect). Abbreviations: RR: Relative risk. CI: Confidence interval. SD: Standard deviation. SOFA: Sequential Organ Failure Assessment. CHF: congestive heart failure. COPD: Chronic obstructive pulmonary disease. CKD: Chronic kidney disease. BMI: Body Mass Index. HIV: human immunodeficiency virus. NIVM: Non-Invasive Mechanical Ventilation. IMV: Invasive Mechanical Ventilation. ICU: Intensive Care Unit. ECMO: Extracorporeal membrane oxygenation. CRRT: Continuous renal replacement therapy. NMB: Neuromuscular blockade. CPR: Cardiopulmonary resuscitation. ICUAP: Intensive Care Unit acquired pneumonia. ARDS: Acute respiratory distress syndrome. * Not shown because of the presence of a zero-cell count in cirrhosis manifested an unbelievably large, estimated coefficient.

## Data Availability

The datasets used and/or analyzed during the current study are available from the corresponding author on reasonable request.

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
