# Peer review of "ICU-Acquired Pneumonia Is Associated with Poor Health Post-COVID-19 Syndrome"

_jcm, 2021, doi:10.3390/jcm11010224_

Round 1

Reviewer 1 Report

In this multicenter prospective study authors evaluate the risk factors associated with development of post covid syndrome or long covid. The idea is interesting and I was very excited to see the results. Unfortunately, after reading the study multiple times I have the following comments , that in my opinion are the major ones:

  1. According to IDSA quidelines pneumonia is classified to community acquired, hospital acquired of ventilator acquired. Intensive care acquired pneumonia as a term doesn't exist to my knowledge. The problem is that this ICU acquired pneumonia, as authors named it- really can be HAP or VAP and authors do not make a distingsion
  2. Symptoms of post COVID syndrome such as numbness and headache are very poorly described. How frequent headaches? Numbness where? etc. This is methodologically extremely poorly executed
  3. Authors failed to mention if these symptoms of post covid syndrome such as fatigue, weakness, astenia have been ruled out before patients were involved in the study. Some patients might have had these symptoms before contracting covid for various reasons- this is a major cofounder
  4. 264 patients were lost in 3 months follow up- what happened to them?
  5. math doesnt add up. There are so many discrepancies in terms of statistics. I will list some: A) you mention that COPD had 100 patients or 8% , this would meen that your sample for 100% would be 1250 which is inaccurate based on Figure 1. B) Female sex was 405 patients or 32.3%- in this case samle would be 1253 patients which again is not accurate as you stated that 991 patient was evaluate in 3 months post discharge. C) NIV in Table 1 was 440, iMV was 858- again 1298 is way more than what you reported in Figure 1. All these make me very uncomfortable that you properly run statistics 
  6. How did you determine that patient had superimposed secondary bacterial or fungal infection? How many patients had positive sputum or blood cultures? Was procalcitonin checked. In my opinion, there is no way you could tell if worsening infiltrates on CXR associated with leukocytosis were due to bacterial/fungal infection or worsening of COVID and leukocytosis was secondary to steroids
  7. LFTs should be replaced with PFTs- L might mean liver while P is pulmonary. This is very confusing
  8. Authors mention about risk factors gotten from multivariate analysis- female sex, duration of ICU stay and onset of " secondary pneumonia " or ICUAP- however, they do not provide cut off values for these two events- consequently I find it useless
  9. Table 5- discrepancy in reported numbers persist- for example oxygen requirement, IMV, NMB- all these numbers yield different sample size and really doesn't project confidence that authors did proper analysis
  10. CXR, CT interpretation- who did it? it is very subjective and authors should have used 2 independent radiologist and have kappa coeficient calculated for interrater agreement

All in all, I have major reservation in validity of this study. I am so sorry that I cannot be more positive. This manuscript has many data collected but unfortunately they were not properly analysed and the entire paper is very confusing.

I would suggest authors to eliminate aim of risk factors associated with long covid and just focus to describe patients with long covid. In that case manuscript might have better chances. 

Author Response

In this multicenter prospective study authors evaluate the risk factors associated with development of post covid syndrome or long covid. The idea is interesting and I was very excited to see the results. Unfortunately, after reading the study multiple times I have the following comments , that in my opinion are the major ones:

According to IDSA quidelines pneumonia is classified to community acquired, hospital acquired of ventilator acquired. Intensive care acquired pneumonia as a term doesn't exist to my knowledge. The problem is that this ICU acquired pneumonia, as authors named it- really can be HAP or VAP and authors do not make a distingsion

Response: We acknowledged that this is a term not included in the IDSA guidelines however the term is valid and extensively reported in critical care. The fist manuscript published was published in 1996 in Intensive Care Medicine. Just in PubMed, there are 15 manuscripts published in the last 10 years referring to ICUAP.

Symptoms of post COVID syndrome such as numbness and headache are very poorly described. How frequent headaches? Numbness where? etc. This is methodologically extremely poorly executed

Responses: The database captured in a follow up visit if the patient had either of this symptoms and as it is reported in Figure one. There was a lack of some granular analysis of some symptoms however the database did not capture the frequency of headaches and/or the location of numbness

Authors failed to mention if these symptoms of post covid syndrome such as fatigue, weakness, astenia have been ruled out before patients were involved in the study. Some patients might have had these symptoms before contracting covid for various reasons- this is a major cofounder

Responses: This is a very important comment and the objective of the study was a voluntary report of  symptoms after the onset of COVID. We have included this clarification in methods and we have made a comment as a potential limitation.

264 patients were lost in 3 months follow up- what happened to them?

Response: Patients did not show up to their appointments despite multiple attempts.  

Math doesnt add up. There are so many discrepancies in terms of statistics. I will list some: A) you mention that COPD had 100 patients or 8% , this would meen that your sample for 100% would be 1250 which is inaccurate based on Figure 1. B) Female sex was 405 patients or 32.3%- in this case samle would be 1253 patients which again is not accurate as you stated that 991 patient was evaluate in 3 months post discharge. C) NIV in Table 1 was 440, iMV was 858- again 1298 is way more than what you reported in Figure 1. All these make me very uncomfortable that you properly run statistics 

Response: Reviewer is right and we want to apologise. The problem is that the percentage was not from the 991 as the final population and we agree this was extremely confusing. We have recalculated all the numbers and included that the target population was only those who were followed up.

How did you determine that patient had superimposed secondary bacterial or fungal infection? How many patients had positive sputum or blood cultures? Was procalcitonin checked. In my opinion, there is no way you could tell if worsening infiltrates on CXR associated with leukocytosis were due to bacterial/fungal infection or worsening of COVID and leukocytosis was secondary to steroids

Response: Diagnosis of ICUAP comprised new or progressive radiologic pulmonary infiltrates together with at least two of the subsequent characteristics: temperature >38ºC or <36ºC; leucocytosis >12,000/mm3 or leucopoenia <4,000/mm3; or purulent respiratory secretion. This is based on published definitions as cited elsewhere. As incorporated in the last paragraph in results, 282 patients had a confirmed pathogen and 220 were VAP. From those, 44 (20%) had early VAP and 176 (80%) had late VAP. Gram negatives represented most pathogens isolated, but S. aureus was present in 46 (16%) cases

LFTs should be replaced with PFTs- L might mean liver while P is pulmonary. This is very confusing

Response: Changed to PFTs

Authors mention about risk factors gotten from multivariate analysis- female sex, duration of ICU stay and onset of " secondary pneumonia " or ICUAP- however, they do not provide cut off values for these two events- consequently I find it useless

Response: Following this reviewer and below’s reviewer recommendation, to explore the factors associated with post-covid syndrome, a generalized linear model was used*, defined by a binomial probability distribution and a log link function. The following variables were included in the multivariable model based on clinical relevance only: sex, SOFA, cirrhosis, non-invasive mechanical ventilation, invasive mechanical ventilation, ICU length of stay, corticosteroids, ICUAP, and ARDS. Relative risks (RRs) and their 95% confidence intervals were calculated. We performed a mixed-effects model* for sensitivity analysis, as defined by a Poisson probability distribution and a log link function, with centres as a random effect, and with an unstructured covariance matrix. All this information is included in methods. We have also included the two continuous variables ( SOFA and ICU-LOS) as dichotomic variables as requested. 

Table 5- discrepancy in reported numbers persist- for example oxygen requirement, IMV, NMB- all these numbers yield different sample size and really doesn't project confidence that authors did proper analysis

Response: We agree with this comment. As we selected the population with 991 patients, there was a discrepancy in table 1, the figure and table 5. Now it’s all corrected and only including data from the patients that were followed up(n=991).

CXR, CT interpretation- who did it? it is very subjective and authors should have used 2 independent radiologist and have kappa coeficient calculated for interrater agreement

Reponses: The radiology interpretation was performed by an independent radiologist in all the cases as per current clinical practice in the participating hospitals. We have included this information in methods. Thank you.

All in all, I have major reservation in validity of this study. I am so sorry that I cannot be more positive. This manuscript has many data collected but unfortunately they were not properly analysed and the entire paper is very confusing.

Response: We are extremely thankful to this reviewer for his/her meaningful analysis and comments. We hope to have provided provide a better version of the manuscript.

I would suggest authors to eliminate aim of risk factors associated with long covid and just focus to describe patients with long covid. In that case manuscript might have better chances. 

Response: We have modified the tables based on the excellent comments and we have just focused on patients with long COVID.

In the present study, Martin-Loeches et al. report that a subgroup of patients with COVID-19 develop persistent COVID-19 symptoms. Furthermore, some patients have very often intensive care unit-acquired pneumonia (ICUAP). A total number of 1,255 ICU patients were scheduled to be followed up at 3 months, the final cohort included 991 (78.9%) patients. A total of 315 patients developed ICUAP, and patients requiring invasive mechanical ventilation had persistent, post-COVID-19 symptoms. The authors identified that female sex, duration of ICU stay, and development of ICUAP were independent risk factors for persistent poor health post-COVID-19. Overall, this study adds important information in this ongoing pandemic in a considerable cohort of patients.

Response: Thanks for the detailed review of the manuscript and the comments.

Martin-Loeches et al reported a cohort study finding on factors associated with poor health post covid. This is a very important topic and will be of interest to JCM readers. The authors might consider the following comment for improvement of the manuscript 

1-Line 106, please add the full stop mark before « additional »

Response: Done

2-Line 133, please add cohort to the design of the study

Response: Added

3-Line 134, Please be more specific about the period of the study from 16 February 2020 to----?

Response: The date has been included  

Outcome

4-The measure of frequency in this study should be incidence, not prevalence (since you considered only the new case of poor health post-covid-19)

Response: Incidence has been corrected as adequately raised in this comment.

Statistical analysis

5-This is a prospective cohort study and the Outcome= incidence of poor health post-COVID-19 in critically ill patients and risk factors. Therefore, the measure of association should be the relative risk, not the odd ratio. 

Response: We agree and all the analysis has been changed as per reviewer’s comments. Now in methods is included “To explore the factors associated with post-covid syndrome, a generalized linear model was used*, defined by a binomial probability distribution and a log link function. The following variables were included in the multivariable model based on clinical relevance only: sex, SOFA, cirrhosis, non-invasive mechanical ventilation, invasive mechanical ventilation, ICU length of stay, corticosteroids, ICUAP, and ARDS. Relative risks (RRs) and their 95% confidence intervals were calculated. We performed a mixed-effects model* for sensitivity analysis, as defined by a Poisson probability distribution and a log link function, with centres as a random effect, and with an unstructured covariance matrix.” 

Please check and use appropriate multivariable analysis test

Response:Please see our previous response as per reviewer’s request

6-In epidemiology, the risk factor needs strong association including causality association proof.

This was not the case of this study and the data could not prove that.

Please revise  the title, objective, and the text by replacing "risk factor" with "factor associated with "

Response: Factor associated with has been included throughout the text

7- Authors' contributions: This is too vague. Please add other contributions in the manuscript( data analyzing, first draft writing, etc...)

Response: These points have been added.

Reviewer 2 Report

In the present study, Martin-Loeches et al. report that a subgroup of patients with COVID-19 develop persistent COVID-19 symptoms. Furthermore, some patients have very often intensive care unit-acquired pneumonia (ICUAP). A total number of 1,255 ICU patients were scheduled to be followed up at 3 months, the final cohort included 991 (78.9%) patients. A total of 315 patients developed ICUAP, and patients requiring invasive mechanical ventilation had persistent, post-COVID-19 symptoms. The authors identified that female sex, duration of ICU stay, and development of ICUAP were independent risk factors for persistent poor health post-COVID-19. Overall, this study adds important information in this ongoing pandemic in a considerable cohort of patients.

Author Response

(The authors gave the same response as above.)

Reviewer 3 Report

 Martin-Loeches et al reported a cohort study finding on factors associated with poor health post covid. This is a very important topic and will be of interest to JCM readers. The authors might consider the following comment for improvement of the manuscript 

1-Line 106, please add the full stop mark before « additional »

2-Line 133, please add cohort to the design of the study

3-Line 134, Please be more specific about the period of the study from 16 February 2020 to----?

Outcome

4-The measure of frequency in this study should be incidence, not prevalence (since you considered only the new case of poor health post-covid-19)

Statistical analysis

5-This is a prospective cohort study and the Outcome= incidence of poor health post-COVID-19 in critically ill patients and risk factors. Therefore, the measure of association should be the relative risk, not the odd ratio. 

Please check and use appropriate multivariable analysis test

6-In epidemiology, the risk factor needs strong association including causality association proof.

This was not the case of this study and the data could not prove that.

Please revise  the title, objective, and the text by replacing "risk factor" with "factor associated with "

7- Authors' contributions: This is too vague. Please add other contributions in the manuscript( data analyzing, first draft writing, etc...)

Author Response

(The authors gave the same response as above.)

Reviewer 4 Report

The authors have carried out a prospective, multicentre and observational  study tittled ICU-acquired pneumonia is a risk factor of a poor health post- Covid-19 syndrome at 40 selected ICUs analizing the poor health post-COVID-19 in the initial cases of hospitalised patients with COVID-19 at 3-month follow-up since hospital discharge.

I highlight some considerations to be taken into account for your knowledge:

  • Institutional affiliations should be homogenous in their content and in the language used. English is recommended.
  • It would be desirable to know the patient characteristics of the population who did not had persistent post-COVID-19 symptoms at 3-month follow-up and see if it is well balanced with the patient characteristics of the population with persistent post-COVID-19 symptoms at 3-month follow-up. We could see if there are significant differences between both populations because that is the basis to fully interpret further results.
  • The sample size is very high (n= 991 patients) but the loss of a 3-month follow-up of 264 patients is striking. How authors justify this loss? It is interesting that it is reflected in the manuscript.
  • Please remove the parentheses below from most of the variables shown in table 1
  • The acronym for Non-invasive mechanical ventilation (NIVM) is misspelled in Table 1
  • Line 404: replace the word fat with at
  • Reference num 17 is incomplete (Line 489)
  • Authors identification are misspelled in reference number 28 (Line 28)

Kind regards

Author Response

In this multicenter prospective study authors evaluate the risk factors associated with development of post covid syndrome or long covid. The idea is interesting and I was very excited to see the results. Unfortunately, after reading the study multiple times I have the following comments , that in my opinion are the major ones:

According to IDSA quidelines pneumonia is classified to community acquired, hospital acquired of ventilator acquired. Intensive care acquired pneumonia as a term doesn't exist to my knowledge. The problem is that this ICU acquired pneumonia, as authors named it- really can be HAP or VAP and authors do not make a distingsion

Response: We acknowledged that this is a term not included in the IDSA guidelines however the term is valid and extensively reported in critical care. The fist manuscript published was published in 1996 in Intensive Care Medicine. Just in PubMed, there are 15 manuscripts published in the last 10 years referring to ICUAP.

Symptoms of post COVID syndrome such as numbness and headache are very poorly described. How frequent headaches? Numbness where? etc. This is methodologically extremely poorly executed

Responses: The database captured in a follow up visit if the patient had either of this symptoms and as it is reported in Figure one. There was a lack of some granular analysis of some symptoms however the database did not capture the frequency of headaches and/or the location of numbness

Authors failed to mention if these symptoms of post covid syndrome such as fatigue, weakness, astenia have been ruled out before patients were involved in the study. Some patients might have had these symptoms before contracting covid for various reasons- this is a major cofounder

Responses: This is a very important comment and the objective of the study was a voluntary report of  symptoms after the onset of COVID. We have included this clarification in methods and we have made a comment as a potential limitation.

264 patients were lost in 3 months follow up- what happened to them?

Patients did not show up to their appointments despite multiple attempts. The information is included in results  

Math doesnt add up. There are so many discrepancies in terms of statistics. I will list some: A) you mention that COPD had 100 patients or 8% , this would meen that your sample for 100% would be 1250 which is inaccurate based on Figure 1. B) Female sex was 405 patients or 32.3%- in this case samle would be 1253 patients which again is not accurate as you stated that 991 patient was evaluate in 3 months post discharge. C) NIV in Table 1 was 440, iMV was 858- again 1298 is way more than what you reported in Figure 1. All these make me very uncomfortable that you properly run statistics 

Response: Reviewer is right and we want to apologise. The problem is that the percentage was not from the 991 as the final population and we agree this was extremely confusing. We have recalculated all the numbers and included that the target population was only those who were followed up.

How did you determine that patient had superimposed secondary bacterial or fungal infection? How many patients had positive sputum or blood cultures? Was procalcitonin checked. In my opinion, there is no way you could tell if worsening infiltrates on CXR associated with leukocytosis were due to bacterial/fungal infection or worsening of COVID and leukocytosis was secondary to steroids

Response: Diagnosis of ICUAP comprised new or progressive radiologic pulmonary infiltrates together with at least two of the subsequent characteristics: temperature >38ºC or <36ºC; leucocytosis >12,000/mm3 or leucopoenia <4,000/mm3; or purulent respiratory secretion. This is based on published definitions as cited elsewhere. As incorporated in the last paragraph in results, 282 patients had a confirmed pathogen and 220 were VAP. From those, 44 (20%) had early VAP and 176 (80%) had late VAP. Gram negatives represented most pathogens isolated, but S. aureus was present in 46 (16%) cases

LFTs should be replaced with PFTs- L might mean liver while P is pulmonary. This is very confusing

Response: Changed to PFTs

Authors mention about risk factors gotten from multivariate analysis- female sex, duration of ICU stay and onset of " secondary pneumonia " or ICUAP- however, they do not provide cut off values for these two events- consequently I find it useless

Response: Following this reviewer and below’s reviewer recommendation, to explore the factors associated with post-covid syndrome, a generalized linear model was used*, defined by a binomial probability distribution and a log link function. The following variables were included in the multivariable model based on clinical relevance only: sex, SOFA, cirrhosis, non-invasive mechanical ventilation, invasive mechanical ventilation, ICU length of stay, corticosteroids, ICUAP, and ARDS. Relative risks (RRs) and their 95% confidence intervals were calculated. We performed a mixed-effects model* for sensitivity analysis, as defined by a Poisson probability distribution and a log link function, with centres as a random effect, and with an unstructured covariance matrix. All this information is included in methods. We have also included the two continuous variables ( SOFA and ICU-LOS) as dichotomic variables as requested. 

Table 5- discrepancy in reported numbers persist- for example oxygen requirement, IMV, NMB- all these numbers yield different sample size and really doesn't project confidence that authors did proper analysis

Response: We agree with this comment. As we selected the population with 991 patients, there was a discrepancy in table 1, the figure and table 5. Now it’s all corrected and only including data from the patients that were followed up(n=991).

CXR, CT interpretation- who did it? it is very subjective and authors should have used 2 independent radiologist and have kappa coeficient calculated for interrater agreement

Reponses: The radiology interpretation was performed by an independent radiologist in all the cases as per current clinical practice in the participating hospitals. We have included this information in methods. Thank you.

All in all, I have major reservation in validity of this study. I am so sorry that I cannot be more positive. This manuscript has many data collected but unfortunately they were not properly analysed and the entire paper is very confusing.

Response: We are extremely thankful to this reviewer for his/her meaningful analysis and comments. We hope to have provided provide a better version of the manuscript.

I would suggest authors to eliminate aim of risk factors associated with long covid and just focus to describe patients with long covid. In that case manuscript might have better chances. 

Response: We have modified the tables based on the excellent comments and we have just focused on patients with long COVID.

In the present study, Martin-Loeches et al. report that a subgroup of patients with COVID-19 develop persistent COVID-19 symptoms. Furthermore, some patients have very often intensive care unit-acquired pneumonia (ICUAP). A total number of 1,255 ICU patients were scheduled to be followed up at 3 months, the final cohort included 991 (78.9%) patients. A total of 315 patients developed ICUAP, and patients requiring invasive mechanical ventilation had persistent, post-COVID-19 symptoms. The authors identified that female sex, duration of ICU stay, and development of ICUAP were independent risk factors for persistent poor health post-COVID-19. Overall, this study adds important information in this ongoing pandemic in a considerable cohort of patients.

Response: Thanks for the detailed review of the manuscript and the comments.

Martin-Loeches et al reported a cohort study finding on factors associated with poor health post covid. This is a very important topic and will be of interest to JCM readers. The authors might consider the following comment for improvement of the manuscript 

1-Line 106, please add the full stop mark before « additional »

Response: Done

2-Line 133, please add cohort to the design of the study

Response: Added

3-Line 134, Please be more specific about the period of the study from 16 February 2020 to----?

Response: The date has been included  

Outcome

4-The measure of frequency in this study should be incidence, not prevalence (since you considered only the new case of poor health post-covid-19)

Response: Incidence has been corrected as adequately raised in this comment.

Statistical analysis

5-This is a prospective cohort study and the Outcome= incidence of poor health post-COVID-19 in critically ill patients and risk factors. Therefore, the measure of association should be the relative risk, not the odd ratio. 

Response: We agree and all the analysis has been changed as per reviewer’s comments. Now in methods is included “To explore the factors associated with post-covid syndrome, a generalized linear model was used*, defined by a binomial probability distribution and a log link function. The following variables were included in the multivariable model based on clinical relevance only: sex, SOFA, cirrhosis, non-invasive mechanical ventilation, invasive mechanical ventilation, ICU length of stay, corticosteroids, ICUAP, and ARDS. Relative risks (RRs) and their 95% confidence intervals were calculated. We performed a mixed-effects model* for sensitivity analysis, as defined by a Poisson probability distribution and a log link function, with centres as a random effect, and with an unstructured covariance matrix.” 

Please check and use appropriate multivariable analysis test

Response:Please see our previous response as per reviewer’s request

6-In epidemiology, the risk factor needs strong association including causality association proof.

This was not the case of this study and the data could not prove that.

Please revise  the title, objective, and the text by replacing "risk factor" with "factor associated with "

Response: Factor associated with has been included throughout the text

7- Authors' contributions: This is too vague. Please add other contributions in the manuscript( data analyzing, first draft writing, etc...)

Response: These points have been added.

Institutional affiliations should be homogenous in their content and in the language used.

English is recommended.

Response: We have made the hospital name in Spanish but the city and country in English

It would be desirable to know the patient characteristics of the population who did not had persistent post-COVID-19 symptoms at 3-month follow-up and see if it is well balanced with the patient characteristics of the population with persistent post-COVID-19 symptoms at 3-month follow-up. We could see if there are significant differences between both populations because that is the basis to fully interpret further results.

Response: We have answered this question to the previous reviewer due to RR instead of the OR and redone the analysis

The sample size is very high (n= 991 patients) but the loss of a 3-month follow-up of 264 patients is striking. How authors justify this loss? It is interesting that it is reflected in the manuscript.

Response: Patients did not show up to their appointments despite multiple attempts. The information is included in results  

Please remove the parentheses below from most of the variables shown in table 1

Response: Done

The acronym for Non-invasive mechanical ventilation (NIVM) is misspelled in Table 1

Response :Done

Line 404: replace the word fat with at

Response: Done

Reference num 17 is incomplete (Line 489)

Response: The ARDS Definition Task Force* author has been added

Authors identification are misspelled in reference number 28 (Line 28)

Response:Done

Round 2

Reviewer 1 Report

The authors have improved the manuscript somewhat and have answered to the concerns that I had. 

Additional comments:

providing the reference for the term ICUAP would be useful in introduction

Table 1- COP( D) is missing

Line 411- it is unclear to this reviewer what this means. Onset after what period of time? Is poor post covid health associated with early onset of icu pneumonia or later and what was the cut off?

Author Response

providing the reference for the term ICUAP would be useful in introduction

Response: ICUAP has been further explained and added a new reference

Table 1- COP( D) is missing

Response COPD and COP were two different acronyms and they have been further included in the table legend. 

Line 411- it is unclear to this reviewer what this means. Onset after what period of time? Is poor post covid health associated with early onset of icu pneumonia or later and what was the cut off?

Response: We found that having a pneumonia during the ICU stay was be a risk factor for recovery. We have  therefore used the word development to clarify the concept, Thank you. 

Reviewer 3 Report

The manuscrit has been improved.

The revised version is fine. My comments have been taken into consideration.

However, the title should be revised accordingly.

The following title could be considered by the authors: ICU-acquired pneumonia is  associated with  a poor health post- 2 Covid-19 syndrome

Author Response

The manuscrit has been improved.

The revised version is fine. My comments have been taken into consideration.

However, the title should be revised accordingly.

The following title could be considered by the authors: ICU-acquired pneumonia is  associated with  a poor health post- 2 Covid-19 syndrome

Response: the title of the manuscript has been included as suggested. 

Reviewer 4 Report

Dear authors,

Thank you very much for reviewing, clarifying and modifying the allegations made previously. It is a very constructive work with high scientific rigor. I have communicated it to the editors of the journal.

Kind regards

Author Response

Thank you very much for reviewing, clarifying and modifying the allegations made previously. It is a very constructive work with high scientific rigor. I have communicated it to the editors of the journal

Response . Thanks